# Position: LLM-Based Social Simulations Require a Boundary

**Zengqing Wu** [1]  **Run Peng** [2]  **Takayuki Ito** [3]  **Makoto Onizuka** [1]  **Chuan Xiao** [1] [4]

## Abstract

This position paper argues that **LLM-based social simulations require clear boundaries to make meaningful contributions to social science**. While Large Language Models (LLMs) offer promising capabilities for simulating human behavior, their tendency to produce homogeneous outputs, acting as an "average persona", fundamentally limits their ability to capture the behavioral diversity essential for complex social dynamics. We examine why heterogeneity matters for social simulations and how current LLMs fall short, analyzing the relationship between mean alignment and variance in LLM-generated behaviors. Through a systematic review of representative studies, we find that validation practices often fail to match the heterogeneity requirements of research questions: while most papers include ground truth comparisons, fewer than half explicitly assess behavioral variance, and most that do report lower variance than human populations. We propose that researchers should: (1) match validation depth to the heterogeneity demands of their research questions, (2) explicitly report variance alongside mean alignment, and (3) constrain claims to collective-level qualitative patterns when variance is insufficient. Rather than dismissing LLM-based simulations, we advocate for a boundary-aware approach that ensures these methods contribute genuine insights to social science.

## 1. Introduction

Social simulation is a modeling tool that employs computational methods to understand social phenomena. Computational methods, particularly those modeling interactions between individuals, demonstrate advantages in capturing the complex and nonlinear behaviors typically inherent in social phenomena (Eidelson, 1997; Remondino

et al., 2010; San Miguel et al., 2012). Among these, Agent-Based Modeling (ABM) is a widely used technique in this area, simulating how individual behaviors and local rules give rise to macro-level patterns (Bonabeau, 2002; Epstein, 1999; Schelling, 1971). ABM offers a bottom-up modeling approach, supports heterogeneity among agents, allows for the exploration of emergent phenomena, and provides researchers with interpretable mechanisms linking micro- and macro-level behaviors (Jackson et al., 2017; Page, 2012; Reeves et al., 2022). Meanwhile, it is controversial due to its reliance on simplification (Edmonds & Moss, 2004), limited adaptability (Wu et al., 2023), sensitivity to initial conditions (Manzo & Matthews, 2014), and challenges in representing subjective or human-like behaviors (Ma et al., 2024; Puig et al., 2021), diminishing the contribution of social simulation methods to social science (Reeves et al., 2022).

Recently, LLM agents and social simulations have attracted growing attention. Existing studies have applied LLM agents to domains such as economics (Han et al., 2023; Li et al., 2024), education (Zhang et al., 2024c), game theory (Sreedhar & Chilton, 2024), and social networks (Wang et al., 2023; Yang et al., 2024c; Zhang et al., 2025a), with claimed advantages like handling natural language, enabling flexible behaviors, and showing human-like reasoning. However, concerns have also been raised: LLMs may carry social and cognitive biases (Mohammadi, 2024; Navigli et al., 2023), lack behavioral diversity (Ma et al., 2025), and are hard to validate or explain (Larooij & Törnberg, 2025; Ma et al., 2024). Whether or not using LLMs is a good protocol for social simulations remains an open question—or may not even be the central question to ask. Many existing studies focus primarily on the simulation itself, while we argue that this narrow focus limits the method's contribution to advancing social science. Before moving forward with more LLM-based social simulations, two critical questions remain:

1. **How can LLM-based social simulations benefit studies of social science?**
2. **Can we draw a line to identify what types of problems are suitable for LLM-based simulations to solve?**

Throughout this paper, we use "boundary" to refer to the *methodological scope conditions* of LLM-based social simulations: the limits of what research claims can be reliably supported given current LLM capabilities, particularly in generating behaviorally diverse agent populations. This usage is distinct from other senses of "boundary", such as

---
[1]University of Osaka [2]University of Michigan [3]Kyoto University [4]Nagoya University. Correspondence to: Chuan Xiao <chuanx@ist.osaka-u.ac.jp>, Zengqing Wu <zengqing.wu@ist.osaka-u.ac.jp>.

*Proceedings of the 43ʳᵈ International Conference on Machine Learning*, Seoul, South Korea. PMLR 306, 2026. Copyright 2026 by the author(s).

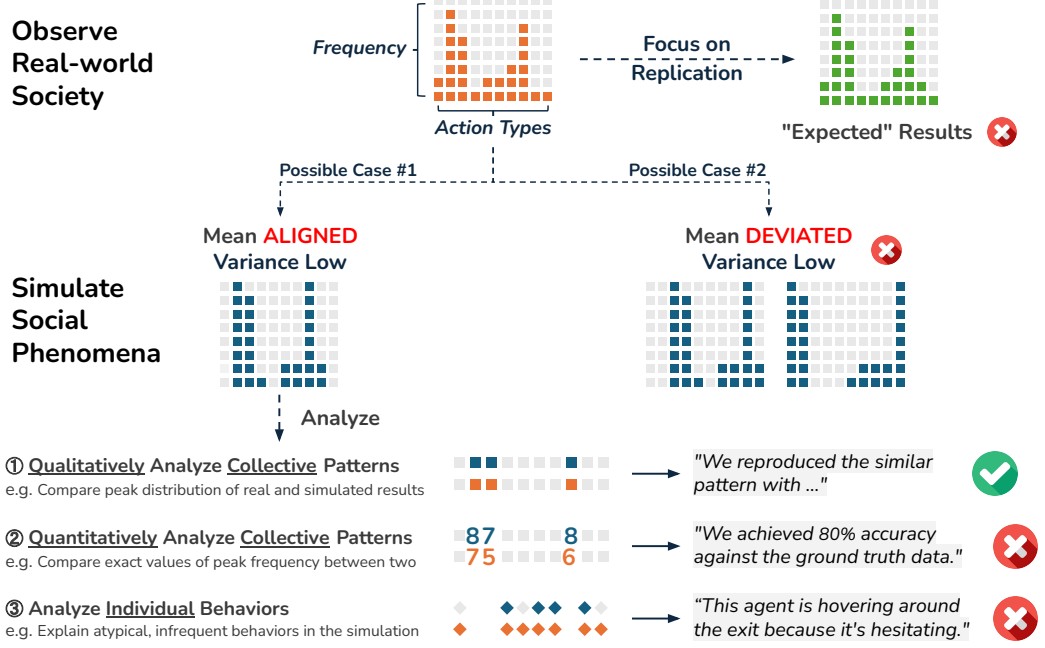

Figure 1. Overview of our claims. We value the goal of social simulations as a means to advance social science, e.g. by explaining social patterns, instead of focusing on "perfect" replication of real-world societies. We further examine possible simulation scenarios (e.g., aligned or misaligned means and variances) and advocate for a stronger emphasis on qualitative analysis of collective patterns.

decision boundaries or safety guardrails, and instead aligns with how validity boundaries are discussed in the social simulation literature (Edmonds et al., 2019).

In this paper, we take the viewpoint that social simulation benefits social science primarily through uncovering social patterns and generating hypotheses. Achieving this requires simulations with sufficient fidelity, particularly in capturing behavioral heterogeneity. We examine how alignment and heterogeneity shape social dynamics, and how the limited behavioral diversity of current LLM agents, i.e., their tendency to act as an "average persona," constrains their effectiveness in representing complex, multi-agent societies (Ma et al., 2025; Shrestha et al., 2024). We analyze common issues in LLM-based simulations through a variance-mean framework (Figure 1) and systematically review current studies to assess current validation practices. We also situate our work within the broader debate by discussing alternative perspectives from optimistic views of LLMs as transformative research tools to skeptical critiques of their fundamental validity. Our central position is that **LLM-based social simulations require clear boundaries, in terms of validation requirements and claim levels, to make meaningful contributions to social science**. We argue this as a general checklist for evaluating the use of LLMs in social simulations, rather than a how-to guide for conducting such studies.

**Our Contributions.** This work makes four key contributions. (1) We systematically analyze the boundary problems of LLM-based social simulations, which are the inherent limitations that fundamentally determine their reliability for social pattern discovery, focusing on the "average persona"

phenomenon where LLMs exhibit insufficient behavioral variance. (2) We discuss simulation fidelity through the concept of agent heterogeneity, indicating why LLMs' tendency to converge towards common patterns fundamentally limits their capacity to simulate complex social dynamics. (3) We conduct a systematic review of 21 recent LLM-based social simulation studies, revealing a gap between the heterogeneity demands of research questions and the depth of validation conducted. While most papers check mean alignment, fewer assess variance, and when they do, LLM behaviors typically show lower diversity than human populations. (4) We provide heuristic boundaries and recommendations for when and how LLM-based simulations can make real contributions to social science research, emphasizing the need to match validation depth to research question requirements. We expect that these boundaries would help bridge the gap between AI and social science communities and contribute to more rigorous findings in social science research.

## 2. LLM-Based Social Simulations

### 2.1. Objectives of Social Simulations

The **primary objective of social simulations** is not to *replicate* reality in fine detail, but to serve as a research tool for explaining social patterns, constructing theories, and providing interpretable foundations for hypothesis generation (Axelrod, 1997; Silverman & Bryden, 2007; 2018). A clear modeling objective is essential for guiding methodological choices. When objectives are poorly defined, effective validation becomes difficult, particularly when

testing alignment with reality and ensuring reproducibility (Arnold, 2014; Axelrod, 1997; Edmonds & Hales, 2003; Edmonds et al., 2019). To clarify the boundaries of social simulation, we examine two objectives frequently declared in LLM-based simulations: replication and prediction. We argue that while both have their place, neither should constitute the primary goal of social simulation.

Replication-oriented work is common in LLM-based simulation literature, while studies achieving novel, valuable social science discoveries through this approach remain limited. Critics note that replication merely repeats known behaviors without revealing new social dynamics or mechanisms (Cheng et al., 2023), which contradicts social simulation's core purpose. Schelling's model exemplifies the alternative (Schelling, 1971): through simple, verifiable interaction rules, it demonstrates universal mechanisms of community segregation without replicating any specific community, revealing broadly applicable social patterns. This suggests that *reproducing* real-world social patterns through simple rules requires no precise *replication* to provide explanatory insights and causal understanding. Furthermore, pursuing exact replication increases parameters and artificial assumptions, risking data overfitting and reducing model verifiability (Larooij & Törnberg, 2025; Silverman & Bryden, 2018). Computational constraints and complexity of sensitivity analysis further obstruct precise replication (Borgonovo et al., 2022; Surve et al., 2023). Hence, social simulations should focus on reproducing and validating key behavioral patterns consistent with real social phenomena (Casti, 1996; Edmonds et al., 2019; Silverman & Bryden, 2018).

Another misconception involves emphasizing *predictive* capabilities through detailed replication performance. Evidence shows limited performance in predicting social dynamics without oracle information, and few effective methods for prediction improvement have been discovered (Gui & Toubia, 2023; Yang et al., 2024a; Ziems et al., 2024). A fundamental concern is that social simulation predictions often constitute mere retrodictions of existing patterns, lacking effective generalization to future scenarios (Edmonds, 2023; Polhill et al., 2021). For instance, using retrodictive tests to claim predictive capabilities (Wang et al., 2025e) may introduce data leakage, as retrospective scenarios could already be contained within the LLM's training data. Such bias is hard to eliminate because LLMs could infer scenarios and implicitly use their knowledge to make "predictions," even when identifying information is removed from prompts (Nguyen et al., 2025a; Zhou et al., 2025). Many simulation works' predictive claims thus exceed actual model capabilities (Ball et al., 2024; Cao et al., 2025; Chuang et al., 2024b; Orlikowski et al., 2025; von der Heyde et al., 2024; Wang et al., 2025c; Yang et al., 2024a; Zhang et al., 2025a), and few studies establish reliable validation methods (Chatterjee et al., 2024). Moreover, some works claim that simulations reflect real social dynamics (Yang et al., 2024c; Zhang et al., 2025a) based on LLMs' explanation of their own decision-making process, which raises endogeneity issues. While

creating comprehensive frameworks for simulating social phenomena at unprecedented scales is valuable, researchers need to be cautious with their objectives and findings.

In sum, social simulation's limitations stem from both LLMs' inherent capabilities and simulation framework design issues (Wang et al., 2025d). We advocate for greater focus on simulation alignment with key social patterns and rigorous validation, rather than treating replication or prediction as core objectives as the foundation of this paper.

### 2.2. Challenges that LLM-Based Simulations Face

We categorize the challenges that LLM-based simulations are now facing into two areas: (1) **usage problems**, which pertain to how researchers apply LLMs and whether these applications align with effective simulation practices; and (2) **boundary problems**, which relate to the inherent capabilities and limitations of LLMs themselves. This paper focuses primarily on the latter. The rationale behind this distinction is to identify the root cause of problems in LLM-based social simulations, specifically, whether they arise from experimental design flaws that can be rectified, or from fundamental issues related to the underlying nature of LLMs.

**Usage Problems**   Usage problems arise from simulation design choices. A common issue is the tendency to aim for perfect replication of reality, which can undermine meaningful social pattern discovery (Edmonds, 2023; Hassan et al., 2013). Other problems include imprecise prompt engineering leading to distortion (Mannekote et al., 2025; Ronanki et al., 2024), overly large action spaces resulting in invalid behaviors (Guo et al., 2024; Liu et al., 2024b;d; Yim et al., 2024), and frameworks introducing excessive researcher assumptions (Silverman & Bryden, 2007). While these usage issues significantly impact simulation effectiveness, they could in principle be mitigated by better practices, and are not the primary focus of this paper.

**Boundary Problems**   Boundary problems represent the inherent limitations of current LLM technology when applied to social simulations. Clarifying these boundaries is essential for understanding where LLM-based simulations can reliably contribute to social science.

Among boundary problems, this paper focuses specifically on the **alignment** problem: whether simulated agents' behaviors and collective dynamics align with real societal patterns. We prioritize alignment because it directly determines whether simulations can genuinely inform our understanding of real social phenomena, i.e., if simulated behaviors systematically diverge from human behaviors, any patterns discovered may reflect LLM artifacts rather than social dynamics. The alignment problem is also closely tied to fundamental characteristics of LLMs, specifically their tendency towards homogeneous outputs that lack the behavioral diversity observed in human populations. In the following sections, we examine why heterogeneity matters

for social simulations (Section 3), how current LLMs fall short in this regard (Section 4), and what this implies for the boundaries of claims that can be reliably made.

We focus on heterogeneity because it directly determines whether simulations can produce the behavioral diversity needed for complex collective dynamics, making it a prerequisite for meaningful social pattern discovery. Beyond alignment, other boundary conditions also affect simulation reliability, including temporal consistency, robustness to perturbations, interaction structure, and environment design. We discuss these additional considerations in Appendix B.

## 3. Alignment and Heterogeneity

The degree of alignment between LLM-based simulations and real-world behavior is key to determining the reliability of insights drawn from social pattern discovery. This alignment can be examined at two levels: **individual-level alignment**, concerning whether each agent behaves in a human-like manner, and **collective-level alignment**, concerning whether agent interactions reproduce realistic social dynamics and emergent phenomena. Understanding the relationship between these two levels is essential before applying LLMs to social simulations.

### 3.1. From Individual to Collective Alignment

While individual-level alignment is often desirable, perfectly capturing individual behavioral patterns is not always essential for social simulations. Social phenomena emerge primarily from interactions between individuals rather than from individual behaviors alone. As Durkheim (2023) argued, collective phenomena possess properties that cannot be reduced to individual psychological states. The emergent properties of social systems cannot be fully predicted from knowledge of individual components alone (Holland, 2000; Louth, 2011; Squazzoni et al., 2014).

Studies in computational social science demonstrate that weak individual alignment can still produce complex collective behaviors. Granovetter (1978)'s threshold models show how simple individual decision rules can produce unpredictable collective outcomes, while Reynolds (1987)' boids model demonstrates how complex flocking behaviors emerge from just three simple rules. An LLM-based simulation reproducing Schelling's model demonstrated that segregated societies emerge even with simple behavior settings and a degree of individual homogeneity (Cheng et al., 2024), illustrating that collective patterns can be relatively insensitive to individual-level modeling imperfections.

However, this does not mean that individual-level characteristics are irrelevant to collective alignment. A crucial distinction must be made: **individual alignment** (whether each agent can exhibit human-like behavior under specific tasks) differs from **heterogeneity** (whether agents differ from each other). While perfect individual alignment may

be unnecessary, collective alignment often depends critically on whether the population exhibits sufficient behavioral diversity (Mou et al., 2024; Lu et al., 2021; Squazzoni et al., 2014). Individual behaviors, through interaction, create feedback loops and emergent effects that constitute collective patterns (Miller & Page, 2009). When agents respond heterogeneously to similar situations, their interactions can produce the non-linear dynamics characteristic of real social systems; when agents respond homogeneously, collective outcomes tend to be predictable (Mondani & Swedberg, 2022).

This insight reframes the question for LLM-based simulations: the key issue is not whether individual agents "pass" as human-like, but whether the *population of agents* exhibits sufficient behavioral diversity to enable realistic collective dynamics. We therefore focus on **output heterogeneity** in this paper's context, i.e., the diversity of behaviors agents actually produced through simulations, rather than input heterogeneity (the diversity of assigned personas), since the setup of diverse personas by means including prompt engineering and fine-tuning do not guarantee diverse behavioral outputs.

### 3.2. Homogeneity and Heterogeneity

**Homogeneity and Its Limitations** Homogeneity, characterized by agents sharing similar behaviors, can in certain cases lead to emergent social patterns. As noted, Schelling's model produces segregation even with uniform agent preferences. However, when agents are highly homogeneous in their decision-making, collective behaviors tend to converge to predictable equilibrium states that can be analytically characterized. In voter models where all agents follow identical imitation rules, the system predictably converges to consensus with mathematically derivable convergence rates (Castellano et al., 2009; Holley & Liggett, 1975). Similarly, in simple contagion models with uniform transmission probabilities, spread patterns follow predictable epidemic trajectories (Hodas & Lerman, 2014; Sprague & House, 2017). Due to this limited complexity, collective behaviors from homogeneous agents can often be characterized through aggregate statistical analyses without complex simulations (Galla et al., 2006; Galstyan et al., 2005; Helfmann et al., 2021). This raises the question: if outcomes from homogeneous agents are analytically tractable, what is the added value of simulations?

**The Critical Role of Heterogeneity** Heterogeneity is widely recognized as a fundamental driver of complex social dynamics and emergent phenomena. Existing works consistently report that certain emergent phenomena only occur with sufficient diversity among agents (Deter & Sayama, 2024; Gao et al., 2024). The importance of heterogeneity has been emphasized across computational simulations, including social network modeling (Ojer et al., 2025), epidemic intervention (Lorig et al., 2021; Reeves et al., 2022), climate policy (Mercure et al., 2016), and wealth formation (Wang et al., 2010), as well as problem-solving applications such

as multi-agent cooperation (Chen et al., 2024) and software development (Hong et al., 2024; Qian et al., 2024).

From a complex systems perspective, when individual differences exist, interactions create feedback mechanisms that amplify these differences, producing emergent phenomena that cannot be predicted from **average** individual characteristics (Miller & Page, 2009). While heterogeneity enables rich interactions that generate intricate patterns (Amin et al., 2018), homogeneity tends to average out behaviors, limiting emergent complexity (Maciejewski et al., 2014). The Condorcet Paradox illustrates this: diverse preferences produce collective voting cycles that cannot be understood by averaging individual preferences (Gehrlein, 1983). Conversely, assuming perfect homogeneity (identical rationality in "Homo economicus") leads to immediate market equilibrium with zero profits, precluding the dynamics that define real economic systems (Grossman & Stiglitz, 1980). We also note that not all research questions require high heterogeneity; when simulations focus on equilibrium existence rather than path dynamics, or on central tendencies rather than distributional properties, requirements may be lower (Appendix E).

### 3.3. Implications for LLM-Based Simulations

These considerations show that neither perfect individual alignment nor homogeneous interactions alone suffice for capturing complex social dynamics. The ability of social simulations to discover novel, complex patterns depends substantially on agent heterogeneity. Whether LLM agent collectives exhibit sufficient heterogeneity therefore becomes a critical indicator of simulation validity. Here, "sufficient" means comparable to the behavioral distribution observed in real human populations, which serves as the **ground truth** for the phenomenon under study. Even when complete ground truth data is unavailable, researchers should compare against whatever empirical benchmarks exist to assess simulation fidelity.

If the phenomena under investigation require heterogeneity for their emergence, but LLMs produce insufficient diversity, conclusions may not reliably apply to real-world situations. The following section examines how heterogeneity is lacking in LLM-based simulations and what this implies for the boundaries of research claims.

## 4. LLM-Based Simulations Lack Heterogeneity

### 4.1. "Average Persona": Origin of Limited Heterogeneity

As established above, sufficient heterogeneity is important for social simulations aiming to reveal complex dynamics. Current LLM agents fall short in generating such diversity. They tend to act as an "average persona," producing responses that reflect population-typical behaviors while suppressing the variation observed in real human populations.

We analyze this limitation through two behavioral dimensions: **variance** (the diversity and spread of behaviors) and **mean** (the central tendency and its alignment with

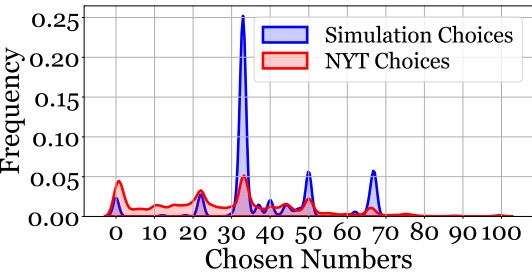

*Figure 2.* Distribution of chosen numbers by GPT-4 (blue) vs. humans (red), adapted from KBC (Wu et al., 2024). The LLM reproduces peak values (33, 50, 66) aligning with human choices, indicating aligned **mean**. However, the frequency of non-peak values is markedly lower than humans, highlighting low **variance**.

real human behaviors). This variance-mean framework helps diagnose alignment problems: variance captures whether LLMs generate sufficient behavioral diversity for complex dynamics, while mean alignment determines whether the central tendency corresponds to real human populations. When characterizing variance as "high" or "low," we refer to comparisons against available ground truth, i.e., empirically observed human behavioral distributions for the phenomenon under study. Even when the data is limited, comparing against existing benchmarks (e.g., from human experiments or surveys) provides a basis for assessing whether LLM-generated behaviors exhibit realistic diversity.

This "average persona" phenomenon stems from training processes. Language model training maximizes conditional probability of predicting text through likelihood-driven loss functions over vast human expression data. This objective inherently rewards high-frequency, mainstream expressions and suppresses marginal ones, fostering an "average persona" that aggregates group thinking and limits distributional representativeness (Nguyen et al., 2025b; Trott, 2024; Wang et al., 2025a). Subgroup heterogeneity is consequently erased, causing behavior to concentrate on dominant patterns that often reflect social biases and demographic stereotypes, even when prompts attempt to elicit alternative perspectives (Liu et al., 2024a; Taubenfeld et al., 2024). This results in difficulty capturing long-tail patterns (Taubenfeld et al., 2024; Wang et al., 2025a). We delineate two primary cases based on variance and mean, each with distinct consequences.

### 4.2. Applicability and Claim Boundaries

**Case 1: Low Variance, Mean Aligned** In this case, LLM agents exhibit low behavioral variance, with strategies and actions concentrated rather than displaying the diversity observed in human populations. However, their mean behavior aligns reasonably well with human averages.

Existing works consistently note that LLMs generate insufficient diversity and exhibit overly homogeneous behavior, often missing human randomness and error patterns (Aher et al., 2023; Anthis et al., 2025; Cheng et al., 2023; Lau et al., 2024). In economic market simulations, while LLM agents replicate macroscopic patterns, they demonstrate significantly less be-

havioral variance than human participants (del Rio-Chanona et al., 2025; Han et al., 2023). In the Keynesian Beauty Contest (KBC, guessing 2/3 of the average), LLM simulations reproduced peak guess values consistent with human experiments, but frequencies on non-peak values were markedly lower (Figure 2) (Wu et al., 2024). In evacuation simulations, despite group-level differences based on personas, individual agent trajectories were surprisingly similar (Wu et al., 2023).

When collective behavioral patterns are meaningful and consistent with real-world outcomes, insufficient variance does not always undermine macroscopic simulation purposes. However, this mandates strict examination of claim boundaries at three levels as shown in Figure 1. **Researchers can focus on collective behavior and qualitative patterns**, as these may be well-reflected despite low individual variance. Conversely, analyzing collective behavior **quantitatively** requires greater caution. As illustrated in Figure 2, the overall distribution shape can differ substantially from human data; claims about precise frequencies, proportions, or distributional statistics may therefore be unreliable even when qualitative patterns align. In addition, interpreting **individual "behavioral trajectories,"** such as specific decisions or paths, can lead to "interpretive overfitting," as individual decisions may not align with reality (Wang et al., 2024a) and are difficult to verify or distinguish from hallucination (Singh et al., 2024). While exploring agent decision logic may enhance AI/ML understanding (e.g., k-level reasoning (Gandhi et al., 2023; Zhang et al., 2025b)), its significance for social science is limited when individual variance is constrained.

**Case 2: Low Variance, Mean Deviated**  The more critical case arises when LLM agents exhibit not only low variance but also mean behavior that deviates significantly from human values, meaning the aggregated LLM behavior does not reflect the central tendency of the targeted human population.

Unlike the Case 1 proposed, where insights into collective patterns might still be obtained, this scenario can render simulations problematic or inapplicable for deriving insights into real human societies. Research finds that LLMs perform significantly differently when simulating population subgroups, often exhibiting biases not present in intended populations (Ma et al., 2025). In public opinion surveys, models trained with human feedback tend towards liberal views and polarized attitudes, difficult to debias through role-play (Bernardelle et al., 2024; Bisbee et al., 2024; Santurkar et al., 2023). Generated dialogues often differ from real conversations in linguistic features (Lin et al., 2024).

Moreover, training processes that debias or rationalize LLM behaviors can paradoxically compromise social simulation utility. When research requires understanding how biases contribute to social patterns, their elimination becomes problematic. Humans exhibit response biases to survey wording that models may not capture (Tjuatja et al., 2024). Cultural deviations are also evident; multilingual simulations show LLM agents making moral judgments inconsistent

with cultural values of those language communities (Jin et al., 2024; Naous & Xu, 2025; Zhang et al., 2024b).

When mean deviation exists, **researchers must check for such deviations**. If the average LLM behavior diverges from actual human population behavior, the simulation's applicability is significantly compromised. Achieving alignment often requires extensive socio-demographic conditions (Argyle et al., 2023; Hu et al., 2025), and reasons for deviations can remain unknown (Nguyen et al., 2025b).

Across both cases, researchers must verify both diversity (variance) and alignment (mean), determining whether observed limitations represent insufficient diversity or deviation from real-world behavior. While various methods have been proposed to enhance agent diversity, significant challenges remain; we discuss existing approaches, their limitations, and potential directions in Appendix G.

## 5. Reviewing Current LLM-Based Social Simulations through the Lens of Our Methodology

To assess how current research aligns with the boundary considerations outlined in previous sections, we reviewed current LLM-based social simulation studies. We selected 21 papers from 2023 to 2025 published at top AI/NLP venues and highly-cited papers that have demonstrated significant influence in the field. Our selection spans diverse domains including economics, social networks, game theory, politics, psychology, and culture. For each paper, we evaluated: (1) the type of research question and its theoretical heterogeneity requirement, (2) whether and how ground truth comparisons were conducted, (3) whether mean alignment and variance were checked, (4) the level of claims made, and (5) whether sensitivity analysis was performed. The detailed criteria for each dimension are provided in Appendix D. Detailed experimental design factors for these selected works are further examined in Appendix C.

### 5.1. Key Observations

**Ground Truth Availability Varies by Domain**  Of the 21 papers reviewed, 14 (67%) included some form of ground truth comparison, while 7 (33%) conducted simulations without human behavioral baselines. Papers studying game theory and economic behavior tend to have better access to ground truth, as these domains have accumulated extensive human experimental data that can serve as benchmarks. In contrast, studies on social network dynamics and cultural phenomena often lack direct human baselines, relying instead on qualitative comparisons with literature or observational patterns. This disparity suggests that certain research domains are currently better suited for validated LLM-based simulations than others.

**Mean Alignment Is More Commonly Checked Than Variance**  Among papers with ground truth, all of them (14 of 14) examined mean alignment, but fewer explicitly assessed

*Table 1.* Systematic review of validation practices in LLM-based social simulation research. **Het. Req.**: Heterogeneity Requirement based on research question type. **GT**: Ground Truth availability and sample size. **Mean**: Whether mean alignment was checked. **Var.**: Whether variance was checked. **Sens.**: Sensitivity analysis conducted. Symbols: ✓ = checked, ✗ = not checked. For Mean/Variance results when checked: Aligned = consistent with human baseline; Deviated/Lower = inconsistent; Mixed = partial alignment. GT Size indicators: (L) = Large, (M) = Medium, (S) = Small. Coll.-Qual. = Collective-Qualitative; Coll.-Quant. = Collective-Quantitative.

| Paper | Domain | Het. Req. | Ground Truth | Mean | Var. | Claim Level | Sens. |
|-------|--------|-----------|--------------|------|------|-------------|-------|
| *High Heterogeneity Requirement* | | | | | | | |
| Lopez-Lira (2025) | Economics | High | None | ✗ | ✗ | Coll.-Qual. | Yes |
| Chuang et al. (2024a) | Social Net. | High | None | ✗ | ✗ | Coll.-Qual. | Yes |
| Wang et al. (2025b) | Social Net. | High | Literature | ✓Align | ✓ | Coll.-Qual. | Part |
| Liu et al. (2024c) | Politics | High | Literature | ✓Align | ✓Low | Coll.-Qual. | Part |
| Hua et al. (2023) | Politics | High | Obs. Data (S) | ✓Mix | ✗ | Coll.-Qual. | Yes |
| Yang et al. (2024c) | Social Net. | High | Obs./Exp. (L) | ✓Mix | ✓Low | Coll.-Qual. | Yes |
| Gao et al. (2023) | Social Net. | High | Obs. Data (L) | ✓Align | ✗ | Coll.-Quant. | No |
| Li et al. (2024) | Economics | High | Obs. Data (M) | ✓Align | ✗ | Coll.-Qual. | Part |
| Tang et al. (2025a) | General | High | Obs. Data (L) | ✓Align | ✓ | Coll.-Quant. | Part |
| *Medium Heterogeneity Requirement* | | | | | | | |
| Huynh et al. (2025) | Game | Med | None | ✗ | ✗ | Coll.-Qual. | Yes |
| Zhang et al. (2023) | Soc. Psych. | Med | None | ✗ | ✗ | Coll.-Qual. | Yes |
| Zhou et al. (2023) | Game | Med | Human Exp. (M) | ✓Dev | ✓Low | Coll.-Qual. | Part |
| Chen et al. (2024) | General | Med | None | ✗ | ✗ | Coll.-Qual. | Part |
| Ren et al. (2024) | Culture | Med | None | ✗ | ✗ | Coll.-Qual. | Part |
| Piatti et al. (2024) | Game | Med | None | ✗ | ✗ | Coll.-Qual. | Yes |
| Xie et al. (2024) | Psychology | Med | Human Exp. (L) | ✓Mix | ✓Low | Coll.-Qual. | Part |
| *Low Heterogeneity Requirement* | | | | | | | |
| Horton et al. (2023) | Economics | Low | Human Exp. (M) | ✓Align | ✓Low | Coll.-Qual. | Part |
| Wu et al. (2024) | Economics | Low | Human Exp. (L) | ✓Align | ✓Low | Coll.-Qual. | Yes |
| Fontana et al. (2025) | Game | Low | Human Exp. (L) | ✓Dev | ✗ | Coll.-Qual. | Yes |
| Akata et al. (2025) | Game | Low | Human Exp. (M) | ✓Mix | ✗ | Coll.-Qual. | Yes |
| Mozikov et al. (2024) | Game | Low | Human Exp. (M) | ✓Mix | ✓Low | Coll.-Quant. | Yes |

variance (9 of 14). This pattern is concerning given our earlier analysis: even when mean behavior aligns with human averages, low variance can fundamentally limit what conclusions can be drawn. Notably, among papers that did check variance, the majority found LLM-generated behaviors exhibited *lower* variance than human populations, which is consistent with the "average persona" phenomenon discussed in Section 4. Only one study clearly reported the relationship between agent scale and variance, with large-scale observational data serving as a reference basis (Tang et al., 2025a). Another study's variance metric was primarily aimed at examining the robustness of the simulation framework, but it was not directly compared with real data (Wang et al., 2025b).

**Heterogeneity Requirements Often Exceed Validation Depth** A notable pattern emerges when comparing research question types with validation practices. Nine papers address research questions with high heterogeneity requirements (e.g., distributional properties, tipping points, path-dependent dynamics), yet only four of these (Wang et al., 2025b; Liu et al., 2024c; Yang et al., 2024c; Tang et al., 2025a) checked both mean and variance against ground truth. Several high-requirement studies (Lopez-Lira, 2025; Chuang et al., 2024a) conducted no ground truth comparison, despite investigating phenomena (market dynamics, opinion polarization) that theoretically depend on behavioral diversity for their emergence. This gap between the

heterogeneity demands of research questions and the depth of validation represents a systematic concern in the field. No work explicitly claimed consistency with human baselines at the variance level in their simulation results. Some works, while having partially comparable results, commendably acknowledged discrepancies between simulation results and real distributions in specific contexts. For example, Liu et al. (2024c) stated that the "belief variance" they tracked could "quickly form a firm opinion" on certain topics such as political issues; Yang et al. (2024c) noted that in their social network simulations, agents were more susceptible to conformity effects than humans. A better practice is Figure 2 of Zhou et al. (2023)'s work, which clearly illustrates the difference of distribution between simulation results and human data. Honest and accurate characterization of this lower variance can actually help readers better identify the paper's useful findings and the boundaries of their applicability.

**Claim Levels Are Generally Appropriate** Encouragingly, most papers (18 of 21) limited their claims to collective-level patterns rather than individual trajectories or quantitative claims, which aligns with our recommendation in Section 4.2. Only three papers made strong collective-quantitative claims (Gao et al., 2023; Tang et al., 2025a; Mozikov et al., 2024), and included ground truth comparisons. This suggests that the community currently maintains a certain level of rigor regarding what claims research can

make at top AI/NLP venues, though the distinction between qualitative and quantitative collective claims deserves more attention in future work. However, a small number of studies potentially overstated their claims. For instance, Lopez-Lira (2025) stated that it "presents a realistic simulated stock market" yet lacks comparison with ground truth. Similarly, Chen et al. (2024) claimed to simulate certain human group behaviors without comparing them with a human baseline. While these works have the potential to contribute to social science discoveries, the claims could be framed with greater rigor.

**Sensitivity Analysis Adoption Is Increasing** The majority of papers (20 of 21) conducted at least partial sensitivity analysis, testing robustness to prompt variations, model choices, or parameter settings. This is a positive trend, as sensitivity analysis helps establish the reliability of simulation findings. However, practices vary considerably: some studies systematically varied multiple factors, while others tested only a single dimension (9 of 21). Standardized sensitivity analysis protocols would benefit the field.

### 5.2. Navigating Contradictory Findings

A critical yet underexplored challenge in LLM-based social simulation research is the existence of contradictory findings across studies examining similar phenomena. These inconsistencies underscore the need for researchers to actively engage with work that may challenge their own conclusions. Here we evaluate two groups of contradictory findings, one in game-theoretic simulations and one in silicon sampling.

**The Cooperation Paradox in Game-Theoretic Simulations** Consider the divergent findings regarding LLM cooperative behavior. Fontana et al. (2025) reported that LLMs such as Llama 2 and GPT-3.5 exhibit hyper cooperative behavior in iterated Prisoner's Dilemma games, forgiving defection rates up to 30% and maintaining cooperation far beyond human baseline levels. They attributed this to an intrinsic preference for positive constructs in these models. However, this finding appears to conflict with Akata et al. (2025), who observed that LLMs can be particularly unforgiving, permanently defecting after experiencing betrayal. Further complicating matters, Piatti et al. (2024) demonstrated in their framework that most LLMs fail to achieve sustainable cooperation in common-pool resource games. Their findings suggest that LLMs exhibit short-sighted, greedy resource extraction patterns rather than the cooperative tendencies reported elsewhere. These contradictions may stem from differences in game structure (dyadic vs. multi-agent), resource framing (abstract payoffs vs. tangible resources), model versions, or prompting strategies. Notably, Fontana et al. (2025) themselves observed that Llama 3 exhibits markedly different, more exploitative behavior, highlighting how rapidly evolving model architectures can invalidate prior behavioral characterizations.

**The Fidelity Paradox in Silicon Sampling** A parallel tension exists in research evaluating LLMs as synthetic survey

respondents. Argyle et al. (2023) introduced the concept of "algorithmic fidelity," defined as the degree to which the complex patterns of relationships between ideas, attitudes, and socio-cultural contexts within a model accurately mirror those within human sub-populations. By conditioning GPT-3 on socio-demographic profiles constructed from thousands of real survey respondents, they reported that the model could reproduce nuanced association patterns, such as relationships between demographics, attitudes, and voting behavior, that closely matched human survey data across multiple subgroups. This concept represents one of the most systematic validity criteria for LLM-based social simulation to date. However, although the original operationalization compared several aggregate distributions and association patterns, it did not make variance matching an explicit or systematic validity requirement. In particular, it remained unclear whether LLM-generated respondents reproduce the within-subgroup dispersion of human responses, rather than merely matching subgroup-conditioned averages or relational patterns. Bisbee et al. (2024) sharpened this concern by showing that, although ChatGPT-generated responses can approximate aggregate opinion averages, they exhibit substantially less variation than ANES responses and often fail to support the same conditional inferences. Their analysis further showed that synthetic data often produced divergent regression-based inferences from the ANES benchmark, sometimes even reversing substantive conclusions, and that the same prompt could yield different response distributions over time, raising concerns about reproducibility. Together, these findings suggest that fidelity in subgroup-conditioned averages or association patterns does not guarantee distributional fidelity, reinforcing the need to validate variance alongside relational alignment.

These contradictions suggest that validation success in one dimension (e.g., mean alignment) may mask failures in another (e.g., distribution collapse). Therefore, we argue that reporting positive validation results is insufficient; researchers must actively investigate potential conflicts with existing literature to define the validity scope of their simulations.

This review reveals both progress and gaps in current validation practices. While the field has developed awareness of ground truth comparison and appropriate claim levels, systematic variance checking remains underutilized, and there is often a mismatch between the heterogeneity demands of research questions and the depth of validation conducted. These findings reinforce our central argument that the value of LLM-based social simulations depends critically on researchers understanding and respecting the boundaries imposed by limited behavioral heterogeneity. We provide detailed recommendations for future validation practices in Appendix F.

## 6. Alternative Views

The rapidly expanding application of LLMs to social simulations has ignited a polarized debate regarding their validity and epistemic status. While our work advocates for

a boundary-aware approach, where utility is contingent upon meeting specific fidelity criteria, perspectives in the broader community generally oscillate between viewing LLMs as transformative "silicon subjects" and dismissing them as fundamentally unreliable surrogates.

**The Optimistic Perspective** Efficiency and Emergence Proponents of LLM-based simulations argue that these models offer a transformative solution to the scalability and cost constraints of traditional human subject research. Anthis et al. (2025) asserted that LLM simulations are a "promising research method" capable of overcoming logistical barriers in data collection, provided that challenges such as diversity and bias are managed. This optimism is shared by Grossmann et al. (2023) and Bail (2024), who envisioned LLMs as powerful instruments for reverse engineering social dynamics and testing interventions in high-stakes environments. Expanding on this potential, Ashery et al. (2025) provided empirical evidence that decentralized LLM populations can autonomously develop social conventions and collective biases. Their findings suggest that these models are not merely stochastic parrots, but are capable of reproducing the spontaneous emergence of complex societal structures without explicit programming, thereby validating the generative potential of AI agents. That said, a question remains: does the emergence of collective conventions in LLM populations necessarily reflect real social dynamics, or could it instead reflect regularities inherited from training distributions? Furthermore, scalability is valuable only insofar as fidelity is maintained; it is worth asking whether simulating large numbers of agents with limited behavioral diversity genuinely advances our understanding of inherently diverse populations.

**The Skeptical Perspective** Fundamental Invalidity and Bias Conversely, a critical body of work questions the fundamental validity of using LLMs as proxies for human behavior. Gao et al. (2025) presented empirical evidence that LLMs consistently fail to replicate human behavior distributions in economic games, warning that their reliance on probabilistic pattern matching lacks the embodied survival objectives that shape human cognition. This skepticism is deeply reinforced by Li et al. (2025a), who characterized current persona-based simulations as a "promise with a catch." Through large-scale experiments, they revealed that prevailing ad hoc generation techniques introduce systematic biases, leading to significant deviations in downstream tasks like election forecasting, and often fail to capture the multi-dimensional attributes of human subjects. Similarly, Larooij & Törnberg (2025) argued that the black-box nature of LLMs may exacerbate rather than resolve historical modeling challenges, making it difficult to disentangle genuine social phenomena from artifacts of the training data. However, wholesale dismissal may overlook evidence that, under appropriate conditions, simulations can reproduce meaningful collective-level patterns (as in the Case 1 analysis in Section 4.2). The challenge may therefore be better characterized as one of conditional validity rather than fundamental invalidity, which is precisely the boundary our framework seeks to establish.

**The Conditional Perspective.** Methodological Rigor as a Prerequisite Bridging these extremes, a growing cohort of researchers aligns with our position that LLM-based simulations are feasible but require strict structural and methodological constraints. Demszky et al. (2023) cautioned that LLMs are "not yet ready" for unsupervised application, a sentiment formalized by Zhou et al. (2025) through the PIMMUR (Profile, Interaction, Memory, Minimal-Control, Unawareness, and Realism) principles, which demonstrate that violations of design standards lead to simulation failure.

Furthermore, Sreedhar et al. (2025) highlighted that validity is often a function of architectural design rather than inherent model capability; they demonstrated that enabling authentic social behaviors (e.g., cheating or cooperation) requires specific simulation mechanisms, such as private communication channels and stake-prompting, without which models fail to capture the complexity of human decision-making. This aligns with Li et al. (2025b) and Kozlowski & Evans (2024), who advocated for cognitively grounded agents and a "validation then simulation" approach. Collectively, these works underscore that valid simulation demands not just behavioral mimicry but the careful engineering of cognitive structures and interaction mechanisms, reinforcing the urgent need for a structured approach to validity. It is within this third, conditional paradigm that we situate our work. By establishing a clear boundary, we aim to move the field beyond the dichotomy of hype and skepticism, ensuring that LLM-based simulations are deployed only where their methodological validity can be rigorously substantiated.

# 7. Conclusion

This paper argues that the primary goal of LLM-based social simulations is to explain social patterns, construct theories, and generate hypotheses. Misunderstandings about these goals in current research have limited their contributions to social science. To better address social science problems, we highlight the need to focus on collective alignment and enhance agent heterogeneity to more accurately reflect real societies. Additionally, other well-established boundaries including individual temporal consistency and simulation robustness are equally essential for applying insights from simulated societies to real-world contexts.

Our core standpoint is to **emphasize the necessity of regulating simulation boundaries, including the scope of claims and simulated problems**. We urge the community to treat these boundaries as **a general checklist for evaluating the use of LLMs in social simulations**, thereby ensuring their **positive contributions to social science research**. Meanwhile, we emphasize advancing the standardization of systematic validation methods for social simulations, as well as enhancing the capability to identify potential biases in simulations, to avoid neglect or bias towards marginalized groups and phenomena.

## Acknowledgements

This work is supported by JSPS Kakenhi JP23K17456, JP23K25157, JP23K28096, JP25H01117, JP26K03246, JST CREST JPMJCR22M2, and JST BOOST JPMJBS2402.

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

# A. Related Works

## A.1. Computational Social Science

Social phenomena typically arise from the interactions of intelligent, adaptive agents under dynamic conditions (Eidelson, 1997; San Miguel et al., 2012). Even when we fully understand behavior at a small scale (e.g., personal behavior), we may not necessarily understand social phenomena at the macro scale (Squazzoni et al., 2014). This complexity presents enormous challenges for social science research, including interpreting causal relationships, determining the applicable scope of problems, and ensuring reproducibility of conclusions. This aligns with sociologist Giddens' proposition that social structures and social practices are interrelated and difficult to find cause-and-effect relationships (Giddens, 1986a;b; Wheeler-Brooks, 2009). Therefore, traditional social science methods, such as surveys and laboratory experiments, struggle to capture the nonlinear and emergent dynamics of real-world social systems. Consequently, they are prone to deriving erroneous patterns from data (known as "apophenia"), and may overlook failure modes not incorporated into the patterns (Abell & Reyniers, 2000; Bragues, 2011; Mondani & Swedberg, 2022). These challenges have driven the rise of computational social science, which attempts to use algorithmic, data-driven, and simulation-based approaches to model and interpret complex social behaviors at scale.

## A.2. Agent-Based Modeling in Computational Social Science

ABM has been a foundational method in computational social science, enabling researchers to simulate macro-level outcomes from simple micro-level behavioral rules (Bonabeau, 2002). Classic examples include Sugarscape (Epstein, 1999) and Schelling's segregation model (Schelling, 1971), which illustrate how wealth gaps or segregation patterns can emerge from individual interactions. Despite disagreements and inconsistencies within social science theories, many works agree that social interaction is the fundamental unit of sociological analysis and plays a crucial role in research, rather than focusing solely on individual behavior or macro structures (Gerring, 2001; Mondani & Swedberg, 2022; Turner, 1988). By ABM, the modeling of social interaction can fill the gap in this micro-macro linkage.

ABM provides explanatory power through controlled simulations, but its limitations are widely acknowledged. These include reliance on hard-coded rules or heuristics, difficulty in encoding subjective behaviors, poor agent adaptability, and simplification of heterogeneity (Edmonds & Moss, 2004; Reeves et al., 2022; Wu et al., 2023). Moreover, the need for handcrafted agent behavior risks introducing researcher bias, and limits the scalability and generalizability of such models to real-world complexity (Williams et al., 2022).

## A.3. LLMs in Social Simulations

Recently, the emergence of LLMs has reignited interest in agent-based simulation by enabling more natural, flexible, and human-like behavioral modeling. LLM agents demonstrate powerful capabilities in understanding ambiguous instructions, simulating subjective decision-making, and generating explanations in natural language (Adornetto et al., 2025; Ma et al., 2024; Park et al., 2023). They show potential across various social science domains: (1) From the technical perspective, LLMs' powerful natural language capabilities and theory of mind (ToM) capabilities expand the boundaries of traditional simulations. For example, the use of LLM agents enables subjective behavioral modeling and the ability to understand ambiguous natural language instructions (Wang et al., 2024b), allows simulation of theory of mind capabilities (Ma et al., 2023), enhances interpretability through generative explanations (Epstein, 2023; Ma et al., 2024), and offers ethical and cost advantages compared to human subject experiments (Mou et al., 2024). (2) From the modeling perspective, LLM agents' generalization capabilities can be leveraged to test various scenarios, creating value across interdisciplinary fields (Mou et al., 2024) and improving the fidelity of complex behaviors such as interaction, collaboration, and gaming (Ma et al., 2024). (3) Exploratory studies have demonstrated human-like behavior, with performance approaching that of humans in certain experiments (Anthis et al., 2025).

However, recent criticisms have highlighted significant limitations. LLM agents may inherit and amplify social biases present in their training data (Ashery et al., 2025; Mohammadi, 2024; Navigli et al., 2023), lack sufficient behavioral heterogeneity (Ma et al., 2025), lack human characteristics such as the ability to learn independently and memory (Ma et al., 2024), and lack transparency and interpretability due to their black-box nature (Larooij & Törnberg, 2025). Furthermore, they tend to collapse to high-probability responses, which limits their ability to simulate the diversity of real human behavior, particularly in contexts with high subjectivity or cultural variability (Shrestha et al., 2024). Validating simulation results and their generalizability to real-world phenomena remains a major open question (Chuang et al., 2024a; Hua et al., 2023; Lorè & Heydari, 2024; Warnakulasuriya et al., 2025). These situations pose challenges in translating the potentials discovered in existing works into findings.

# B. Other Potential Boundaries

Beyond the alignment and heterogeneity issues discussed in the main text, other boundary conditions affect the reliability of LLM-based social simulations. We briefly discuss four additional considerations: temporal consistency, robustness, interaction structure, and environment design.

### B.1. Temporal Consistency

In multi-round social simulations, LLM agents may fail to maintain cognitive consistency in their roles during extended interactions (Huang et al., 2024). Unlike single-round Q&A, long-term simulations require agents to behave consistently over time. However, LLMs lack continuous memory capabilities and respond passively to context. Each API call produces independent responses related only to current input, even when previous actions are reprompted to simulate memory. Slight differences in context across rounds may cause the same agent to produce inconsistent reactions (Yao et al., 2023; Zhu et al., 2023).

When an agent's behavioral traits significantly influence other agents' behaviors, inconsistency-induced trait changes may alter macro patterns in the simulation. Without verification of temporal consistency, researchers might misinterpret pattern changes as emergent phenomena rather than recognizing them as artifacts of LLM limitations.

**Implications for researchers:** For long-term simulations, researchers are suggested to verify that key agent characteristics remain stable across rounds, and distinguish between genuine emergent dynamics and artifacts of persona drift.

### B.2. Robustness

Robustness refers to whether simulation conclusions remain stable and reproducible under different parameter settings, conditions, and perturbations. The difficulty of LLMs in providing repeatable results is a major challenge, and necessary sensitivity analysis practices are rarely implemented (Larooij & Törnberg, 2025).

In LLM-based simulations, robustness is primarily verified through sensitivity analysis: examining whether qualitative patterns are sensitive to minor differences in context or prompts (Hosseini & Horbach, 2023; Yang et al., 2024b; Ziems et al., 2024). LLM sensitivity varies significantly; in some situations, LLMs display excessive sensitivity towards certain groups or topics, while in others they achieve better balance (Zhang et al., 2024a). Whether simulations maintain discovered patterns under perturbations constitutes one boundary of the simulatable range.

**Recommended sensitivity checks:** Researchers should test robustness across multiple dimensions: prompt wording (do minor rephrasings change outcomes?), persona descriptions (are results stable across paraphrased definitions?), initial conditions (do different starting configurations yield consistent patterns?), and model parameters (how do temperature and other settings affect conclusions?).

### B.3. Interaction Structure

The design of interaction mechanisms can substantially affect what social behaviors emerge in simulation. Sreedhar et al. (2025) demonstrated that enabling authentic social behaviors such as cheating or cooperation requires specific simulation mechanisms (e.g., private communication channels and stake-prompting), without which models fail to capture the complexity of human decision-making. This suggests that validity is often a function of architectural design choices in the simulation framework, not solely of the underlying model's capability. Researchers should therefore consider whether their interaction structure (pairwise, group, network) adequately supports the social phenomena under investigation, and report how structural choices may constrain or enable the behaviors observed.

### B.4. Environment Design

The action space and environmental constraints of a simulation shape the range of possible agent behaviors. Overly large action spaces can lead to invalid or implausible behaviors (Guo et al., 2024; Liu et al., 2024b), while overly constrained environments may prevent the emergence of the very phenomena researchers seek to study. Environment design also includes how information is made available to agents (full observability vs. partial), how time progresses (synchronous vs. asynchronous updates), and what feedback agents receive from their actions. These choices interact with LLM capabilities in non-trivial ways and can independently affect simulation outcomes.

*Table 2.* Experimental design factors across the 21 reviewed LLM-based social simulation studies, ordered by heterogeneity requirement (same as Table 1). Persona: D=Demographic, P=Personality-based, DD=Data-driven, M=Minimal/None, Auto.=Automated. Prompting is characterized along two axes: reasoning depth (Zero-shot / CoT / Multi-step) and memory mechanism (None / History / Reflection). Temp.: temperature setting; NR=Not Reported. Var.: variance comparison result from Table 1; Comp.=Comparable to human, Low=Lower than human, –=Not reported.

| Paper | Het. | Persona | Prompting | Model(s) | Temp. | Agents | Interaction | Var. |
|---|---|---|---|---|---|---|---|---|
| *High Heterogeneity Requirement* | | | | | | | | |
| Lopez-Lira (2025) | High | P | CoT; history | GPT-4o | NR | 8–10 | Group | – |
| Chuang et al. (2024a) | High | D | CoT; reflection | GPT-3.5/4; Vicuna | 0.7 | 10–20 | Pairwise | – |
| Wang et al. (2025b) | High | D+P | Multi-step; reflection | GPT-4o-mini | NR | 50 | Network | Comp. |
| Liu et al. (2024c) | High | D+P | Multi-step; reflection | GPT-3.5 | NR | 30 | Network | Low |
| Hua et al. (2023) | High | DD+P | Multi-step; history | GPT-3.5/4; Claude-2 | NR | 7–8 | Group | – |
| Yang et al. (2024c) | High | DD+D+P | CoT; history | Llama3-8B (+others) | NR | ∼1M | Network | Low |
| Gao et al. (2023) | High | DD+D | Zero-shot; history | GPT-3.5; ChatGLM | NR | 8.5K–18K | Network | – |
| Li et al. (2024) | High | D+DD | Multi-step; reflection | GPT-3.5 | NR | 100–300 | Group | – |
| Tang et al. (2025a) | High | D | Zero-shot; history | LLaMA3-8B; GPT-4o | NR | ∼100K | Hybrid | Comp. |
| *Medium Heterogeneity Requirement* | | | | | | | | |
| Huynh et al. (2025) | Med | P | Zero-shot; history | GPT-4o; Claude; Mistral; Llama | NR | 2–3 | Pairwise/Group | – |
| Zhang et al. (2023) | Med | P | Multi-step; reflection | GPT-3.5; LLaMA2; Qwen | 0.0–0.75 | 2–10 | Group | – |
| Zhou et al. (2023) | Med | D+P | Zero-shot; history | GPT-4/3.5; LLaMA2; MPT | 1.0 | 2 | Pairwise | Low |
| Chen et al. (2024) | Med | Auto. | CoT; reflection | GPT-3.5/4 | NR | 2–4 | Group | – |
| Ren et al. (2024) | Med | P | CoT; reflection | GPT-3.5/4 | NR | 10 | Group | – |
| Piatti et al. (2024) | Med | M | CoT; history | 15 models | 0.0 | 5 | Group | – |
| Xie et al. (2024) | Med | D | CoT; reflection | GPT-3.5/4; LLaMA2; Vicuna | 1.0 | 53 | Pairwise | Low |
| *Low Heterogeneity Requirement* | | | | | | | | |
| Horton et al. (2023) | Low | P+D | Zero-shot; none | GPT-3 | NR | ∼100 | Single | Low |
| Wu et al. (2024) | Low | M | Multi-step; history | GPT-4; Claude-3; GPT-3.5 | 0.0–1.2 | 2–400 | Hybrid | Low |
| Fontana et al. (2025) | Low | M | Zero-shot; history | LLaMA2/3-70B; GPT-3.5 | 0.7 | 1 (scripted) | Pairwise | – |
| Akata et al. (2025) | Low | M | CoT; history | GPT-4; davinci; Claude-2; LLaMA2 | 0.0 | 2 | Pairwise | – |
| Mozikov et al. (2024) | Low | M | CoT; history | GPT-3.5/4/4o; Claude; +others | 0.0–1.0 | 2+ (varied) | Pairwise/Group | Low |

## C. Review of Experimental Design Factors of Current Works

Table 2 documents the experimental design choices across the 21 reviewed studies in Section 5. A key observation is that lower-than-human variance appears across highly diverse design configurations. Among the 9 papers that explicitly assessed variance:

- **Persona construction** does not reliably predict variance outcomes. Both comparable-variance studies used demographic-based personas (Wang et al. (2025b): D+P; Tang et al. (2025a): D), but the same persona types also appear in lower-variance studies (e.g., Liu et al. (2024c): D+P; Xie et al. (2024): D with 53 personas). Even the most richly specified construction, which combines data-driven, demographic, and personality-based approaches (Yang et al., 2024c), still produced lower-than-human variance.
- **Model choice** shows no clear association. Lower variance was observed across GPT-3 (Horton et al., 2023), GPT-3.5 (Liu et al., 2024c), GPT-4 (Wu et al., 2024), Llama (Yang et al., 2024c), and multi-model comparisons (Mozikov et al., 2024; Zhou et al., 2023). The two comparable-variance studies used GPT-4o-mini and LLaMA3-8B/GPT-4o, both of which also appeared in lower-variance contexts.
- **Temperature settings** do not resolve the issue. Lower variance persists at both $T=0.0$ (Mozikov et al., 2024) (for its main experiment) and $T=1.0$ (Zhou et al., 2023; Xie et al., 2024), and Wu et al. (2024) explicitly tested 0.0–1.2 while still reporting lower variance.
- **Interaction structure and scale** are similarly inconclusive. Lower variance appears in pairwise (Zhou et al., 2023), network (Liu et al., 2024c; Yang et al., 2024c), hybrid (Wu et al., 2024), and single-agent (Horton et al., 2023) settings, spanning 2 to ∼1M agents.

This cross-cutting pattern suggests that insufficient behavioral heterogeneity is not primarily attributable to any particular design decision, but rather reflects a more fundamental characteristic of current LLMs. This supports the distinction between *usage problems* and *boundary problems* in Section 2.2: while design choices can influence heterogeneity, they do not appear sufficient to overcome the inherent tendency of LLMs toward behavioral homogeneity.

## D. Review Criteria for Systematic Analysis

This appendix details the criteria used to evaluate papers in Table 1.

## D.1. Research Question Type and Heterogeneity Requirement

Our classification draws on established frameworks in computational social science and ABM literature. Following Squazzoni et al. (2014) and Edmonds et al. (2019), we recognize that different modeling purposes impose different requirements on agent heterogeneity. We adapt this idea to categorize research questions by the degree to which behavioral variance (rather than just central tendency) is essential to the phenomenon under study.

We classified research questions into the following types:

- **Equilibrium**: Whether a system can reach a particular stable state.
- **Central Tendency**: How average or typical behavior evolves over time.
- **Distribution**: Properties concerning the full distribution of behaviors (inequality, polarization, and diversity).
- **Tipping Point**: Critical thresholds, phase transitions, or cascade phenomena.
- **Path-dependent**: Outcomes that depend on the sequence or history of actions.

Based on these types, we assigned heterogeneity requirements following the principle that *research questions whose answers depend on distributional properties (not just averages) require higher agent heterogeneity* (Reeves et al., 2022)

- **Low**: Questions primarily concerning equilibrium existence or central tendency dynamics, where the specific distribution shape matters less.
- **Medium**: Questions involving some distributional aspects but where central tendencies remain important.
- **High**: Questions fundamentally concerning distribution shape, tails, subgroup differences, tipping points, or path-dependent dynamics.

We acknowledge that this classification involves judgment calls. The primary criterion was: *"Would the research question still be answerable if all agents behaved identically at the population mean?"* If yes, heterogeneity requirement was coded as Low; if partially, Medium; if no, High. For instance, the primary question in Piatti et al. (2024) is to investigate collective survival phenomena, while this work also measured inequality using the Gini coefficient. In this case, we regard this as a Medium case. Another example is Tang et al. (2025a). It is a High case as the paper indicated "a small number of individuals may lead to very large fluctuations of the simulation results". For borderline cases (e.g., studies that examine both equilibrium existence and distributional properties), heterogeneity requirement was coded based on the primary research question as stated by the authors.

## D.2. Ground Truth Categories

- **Human Experiment**: Controlled experimental data from human participants.
- **Observational Data**: Real-world behavioral records (e.g., social media data and transaction logs).
- **Literature**: Qualitative comparison with established findings in prior research.
- **None**: No human behavioral baseline provided.

Sample size indicators: Large (L) = thousands of data points or participants; Medium (M) = hundreds; Small (S) = fewer than 100.

## D.3. Alignment Assessment

**Mean Alignment** was marked as checked (✓) if the paper explicitly compared average LLM agent behavior against human baselines. Results were categorized as:

- **Aligned**: LLM mean behavior is statistically or qualitatively consistent with human average.
- **Deviated**: LLM mean behavior significantly differs from human average.
- **Mixed**: Alignment varies across conditions or measures.

**Variance** was marked as checked (✓) if the paper explicitly examined the spread or diversity of LLM agent behaviors. If real human data is available for comparison, the results are further classified as:

- **Comparable**: LLM behavioral variance is similar to human population variance.
- **Lower**: LLM behavioral variance is notably less than human population variance.
- **(None)**: The variance of the simulation data was reported, but was not directly compared with the ground truth.

## D.4. Claim Level

- **Individual Trajectory**: Claims about specific agent behaviors or decision paths.

- **Collective-Qualitative**: Claims about qualitative collective patterns or trends.
- **Collective-Quantitative**: Claims about precise quantitative metrics matching human data.

### D.5. Sensitivity Analysis

- **Yes**: Systematic testing across multiple dimensions (prompt variations, model choices, parameters, initial conditions).
- **Partial**: Testing on at least one dimension but not comprehensive.
- **No**: No sensitivity analysis reported.

## E. When Is Heterogeneity Less Critical?

While we emphasize heterogeneity's importance, not all research questions have equal requirements for behavioral diversity.

**Lower Heterogeneity Requirements**   The following research question examples may tolerate lower agent heterogeneity:

- **Equilibrium existence:** When research asks whether a system *can* reach a particular state (e.g., "Can markets clear?"), the specific path or distribution may matter less than the qualitative outcome.
- **Central tendency dynamics:** When studying how average behavior evolves (e.g., "Which direction does mean opinion shift?"), the distribution spread may be less critical.
- **Structural effects:** When outcomes are driven primarily by network structure or spatial arrangement rather than agent diversity.
- **Robustness demonstrations:** When showing that a pattern emerges "even under" simplified conditions, homogeneous agents can serve as a conservative test.

**Higher Heterogeneity Requirements**   Conversely, these research questions fundamentally require heterogeneity and may be unsuitable for current LLM-based simulations:

- **Distributional properties:** Studying inequality, polarization, or outcomes where distribution *shape* matters (not just the mean).
- **Tipping points:** Understanding when systems undergo phase transitions often depends on heterogeneous distributions of thresholds or sensitivities.
- **Path-dependent outcomes:** When the sequence of who acts first matters, heterogeneity in timing and responsiveness is essential.
- **Minority influence:** Studying how small groups or rare behaviors affect collective outcomes requires capturing distribution tails.
- **Subgroup-specific dynamics:** Research targeting specific demographic or social subgroups, especially marginalized groups likely underrepresented in LLM training data.

**Decision Heuristic**   As a practical heuristic: "If I replaced every agent with the population mean, would my research question still be answerable?" If yes, heterogeneity requirements are likely lower. If no, because the question concerns variance, tails, or subgroup differences, current LLM-based simulations may not be appropriate.

## F. Recommendations for Future Research

Based on the observations from our systematic review (Section 5), we offer the following recommendations:

1. **Match validation depth to research questions.** Studies investigating distributional properties, tipping points, or path-dependent phenomena should prioritize variance validation, not just mean alignment.
2. **Report variance explicitly.** Even when variance appears lower than human baselines, documenting this limitation helps readers appropriately scope the findings. Future work should establish threshold values for "acceptable" divergence from human baselines, potentially through community consensus or meta-analytic benchmarks.
3. **Leverage existing human data.** Domains with accumulated experimental data offer more tractable starting points for validated simulation research. Also, initiatives such as Many Labs (Klein et al., 2014) and the Reproducibility Project (Open Science Collaboration, 2015) provide pre-registered, multi-site human data that can serve as robust benchmarks.
4. **Develop domain-specific benchmarks.** For domains where ground truth is difficult to obtain (e.g., large-scale social network dynamics), the community would benefit from shared benchmark datasets and evaluation protocols.
5. **Conduct contradiction analysis.** Researchers may actively search for and document existing studies that report findings

potentially contradicting their own. This "contradiction audit" should examine whether observed LLM behaviors (e.g., cooperation levels, opinion distributions, and decision-making patterns) align with or diverge from related works using different models, prompting strategies, or experimental designs. When contradictions are identified, researchers can explicitly discuss possible sources of divergence, including model version differences, task-framing variations, and temporal instability, rather than treating the findings in isolation. This practice will help establish the boundary conditions under which specific LLM behavioral patterns hold and prevent the spread of incompatible claims in the literature.

# G. Challenges and Directions for Improving Heterogeneity

### G.1. Challenges in Existing Approaches

Various methods aim to construct diverse agents, including prompt engineering (Park et al., 2022), personality-based prompting (Serapio-García et al., 2023), character modeling from interviews (Jung et al., 2025; Park et al., 2024), and large-scale data alignment (Ge et al., 2024; Li et al., 2025c). However, these approaches face limitations. Prompt engineering often cannot eliminate bias, especially for minority groups. Large-scale alignment is costly, with scarce high-quality data and anonymity issues affecting generalization (Li et al., 2025c). As simulation scale increases, detailed modeling costs rise dramatically, forcing trade-offs between precision and scale (Chen et al., 2025; Mou et al., 2024). Multi-LLM approaches still show behaviors concentrating on few strategies (Fontana et al., 2025; Lu, 2024).

Furthermore, individual-level alignment does not guarantee collective alignment. Bias removal may weaken knowledge maintenance and performance (Chen et al., 2025). Standardized methods to confirm which approaches achieve both diverse and aligned heterogeneity remain lacking, and effects of adding personas can be inconsistent across contexts (Zheng et al., 2024). Prompting may capture only superficial personas, struggling to represent deep beliefs and decision-making processes.

### G.2. Potential Directions

Several research directions may help address these limitations, though none has yet been demonstrated to fully resolve them.

From a *data* perspective, curating training corpora that better represent social and cultural diversity, informed by sociolinguistic theories of language variation (Yuan et al., 2024; Li et al., 2025c; Grieve et al., 2025), may help preserve distributional tails that current training objectives suppress. Meta-prompting-driven approaches for generating diverse synthetic data (Riaz et al., 2025) suggest that carefully mixing real-world and synthetic data may mitigate long-tail collapse, though their applicability to social simulation remains to be tested.

From a *feedback* perspective, designing reward signals that value response diversity beyond average preferences could reduce homogenization during alignment (Achintalwar et al., 2024; Liu et al., 2023). *Hybrid aggregation* strategies that combine human and LLM sources show promise for mitigating bias through complementary strengths in accuracy and diversity (Abels & Lenaerts, 2025). *Multi-model ensembles* using multiple LLMs may increase population-level diversity, though existing evidence suggests that even multi-LLM approaches can concentrate on few strategies (Fontana et al., 2025; Lu, 2024).

At the *architecture* level, approaches such as diversity-constrained decoding and Mixture-of-Experts (MoE) with persona-specific routing (Tang et al., 2025b) represent intriguing directions, but remain largely unexplored for social simulation. Recent work continues to identify significant bottlenecks in capturing complex human heterogeneity, cultural depth, and dynamic preferences through current intervention methods (Prama et al., 2025; Venkit et al., 2026), suggesting that a boundary-aware framework remains a necessary complement to technical improvements.

# H. Challenges and Future Directions

The boundaries of LLM-based simulations present several challenges and areas for improvement.

**(1) Validation.** While validation of LLM individual behavior and dynamic interactions is more difficult compared to traditional ABM methods, there is currently a lack of good evaluation methods, with heavy reliance on manual or LLMs' self-report approaches for validation (Adornetto et al., 2025; Mou et al., 2024). In response, the simulation community needs to promote systematic evaluation standards to examine whether LLM-based simulations can yield conclusions beneficial for understanding real society.

**(2) Conditions of claims.** Social simulation research needs to more rigorously consider the proper claims of simulation conclusions, including clearly defining the conditions under which conclusions hold, their scope of applicability, and their generalization ability in real-world contexts, avoiding overclaims that reduce the credibility and applicability of simulation conclusions. For instance, while simulations with constrained heterogeneity can produce findings consistent with general

patterns, as demonstrated by case studies showing that organizational diversity typically does not improve collective performance, researchers must meticulously bound their claims, as these simulations may fail to capture specific conditions (e.g., extreme individual bias) where the opposite effect occurs (Xu et al., 2014), and dramatically increased heterogeneity may reveal emergent phenomena beyond the original scope.

**(3) Bias and ethical concerns.** Close attention needs to be paid to bias issues in LLM-based simulations. Limited by the lack of heterogeneity in LLMs, simulations may lead to neglecting marginalized groups or generating stereotypes and negative biases towards specific populations or phenomena. It is necessary to confirm whether LLMs capture biased "averages" and conduct moral and ethical considerations.

**(4) Empirical research.** Considering that our ultimate goal is to contribute to society, applying findings from social simulations to empirical solutions to real-world problems to confirm or refute the reliability of conclusions may be the next step the community needs to actively take to enhance the credibility and importance of simulation methods in research (Popper, 2005; Watts, 2017).

