# OpenReview forum: "Position: LLM-Based Social Simulations Require a Boundary"
_ICML.cc/2026/Position_Paper_Track — ICML 2026 Position Paper Track regular_

### Official Review · Reviewer_eXdr · 2026-03-08

**Significance:** 2
**Argument Clarity:** 3
**Rating:** 4
**Confidence:** 3

**Questions:**

1. While the paper do suggest actionable suggestions for those who aim to use LLM for simulating social behaviours in their experiments, I am curious if there are actions that can be made from a more technical side, i.e., which certain characteristics or methodologies the community should aim to incorporate into LLMs training to better allow for LLMs that better model social phenomena. I suspect this may relate to points 3 and 4 into the Call for Future Research, but I am also curious if they can be elaborated further.

2. Are there reasons that the community have been focused on individual-level alignments of LLM even when they may really be aiming to get a collective-level understanding of some hypothesis? Is it in terms of technical simplicity that it is easier to align a LLM to one person, or are there some more fundamental assumption that researchers are typically working under that is harmful?

**Alternative Views Section:**

Yes

**Compliance With Llm Reviewing Policy A Conservative:**

Affirmed.

**Discussion Potential:**

2

**Final Justification:**

I am already in favour for acceptance of the paper. The rebuttals from the authors have mainly clarified some bit that I am unsure with, hence an increase in confidence for my rating.

**Paper Summary:**

The paper argues for results based on social simulations by LLMs, and discuss why some of the results may not be so reliable as reported in practice. The paper describes the issue that LLMs typically do not have enough heterogeneity in generated answers to simulate a distribution of people, often resulting in too much similarity (low variance) in responses and sometimes mismatch with the actual mean responses as well. The paper then sets guidelines that papers should report not only the mean results but also the variation and how well it matches the true underlying distribution -- something that isn't currently being done in many published works relying on social simulations.

**Position:**

Yes

**Position In Title:**

Yes

**Related Work:**

3

**Strengths And Weaknesses:**

Strengths:

- The paper provides interesting missing pieces in simulation-based research, and can serve as a good reminder for the community that simulations should not be trusted blindly and results should be provided with appropriate disclaimers (i.e., variance in results).

- The points are well-backed via case studies of existing papers, and it is interesting to see the analysis in how the data in those works are not appropriately reporting the information regarding the uncertainty.

Weaknesses:

- It can be argued that while reporting the variance is important for LLM-based simulation (and in fact for any LLM-based research in general as well), the position doesn't necessarily seem controversial or may create much discussion. I suspect most researchers would agree that reporting variance is important to quantify how much we can trust a piece of work, even if they do not do it themselves (for whatever reason). I am wondering if the position would be more worthy of discussion if it focuses more on when such simulations can be used and provide benefits that outweighs risks, or how the simulations can be adopted to match these expectations.

- The paper has mentioned some alternate views, however it would be interesting as well to discuss why a view that is too extreme in the either direction may not be exactly what the position paper is advocating and why they may be untrue from the authors' point of view (as opposed to just stating what these views are). The alternate views are not tackled with enough criticalness with respect to the authors' position.

**Support:**

3

---

> ### Author Rebuttal · Authors · 2026-03-29
>
> Thank you for your thoughtful and valuable comments.
>
> > W1: The position may not be controversial enough.
>
> We sincerely thank the reviewer for this observation. We agree the abstract principle of reporting variance is uncontroversial.
>
> Our contribution lies in **documenting the gap between principle and practice**: among 14 papers with ground truth, only 9 assessed variance, and the majority found lower-than-human variance. We also identify a **systematic mismatch**: 9 papers have high heterogeneity requirements, yet only 4 checked both mean and variance. Our position extends beyond "report variance" to a **systematic framework**: (1) classifying research questions by heterogeneity requirements, (2) matching validation depth accordingly, and (3) constraining claim levels based on observed variance.
>
> > W2: Alternative views not tackled with enough criticalness.
>
> We fully agree. Our comparative rather than adversarial framing was a deliberate choice, but we will add more pointed critiques:
>
> **Optimistic perspective:** While Anthis et al. (2025) and Ashery et al. (2025) demonstrated LLMs' ability to generate emergent phenomena, this does not guarantee such phenomena reflect real social dynamics rather than training distribution artifacts. Scalability is valuable only if fidelity is maintained; simulating millions of homogeneous agents does not advance understanding of inherently diverse populations.
>
> **Skeptical perspective:** Wholesale dismissal overlooks evidence that, under appropriate conditions, simulations reproduce meaningful collective-level patterns (our Case 1 analysis). The challenge is not fundamental invalidity but conditional validity, which is the boundary that our framework establishes.
>
> We will incorporate these more pointed critiques into Section 6 of the revised manuscript.
>
> > Q1: Technical actions for improving LLM training for social simulation.
>
> We also believe that beyond identifying boundaries, understanding how to expand them matters. Section 4.2 finds current methods effective in specific scenarios (e.g., Park et al. (2024) on interview-based character modeling) while population-level diversity remains open.
>
> We will expand our discussion with some current directions: (1) training data optimized for social/cultural diversity [1–3]; (2) reward signals valuing response diversity [4–5]; (3) hybrid human-LLM aggregation [6]. These are active research areas; comprehensive treatment is beyond this paper's scope, but they merit inclusion as broadly recognized concerns.
>
> [1] Yuan et al. "Measuring social norms of large language models." NAACL 2024 Findings.
>
> [2] Li et al. "From 1,000,000 users to every user: Scaling up personalized preference for user-level alignment." arXiv:2503.15463 (2025).
>
> [3] Grieve et al. "The sociolinguistic foundations of language modeling." Frontiers in Artificial Intelligence 7 (2025): 1472411.
>
> [4] Achintalwar et al. "Alignment studio: Aligning large language models to particular contextual regulations." IEEE Internet Computing 28.5 (2024): 28-36.
>
> [5] Liu et al. "Trustworthy LLMs: a Survey and Guideline for Evaluating Large Language Models' Alignment." SoLaR, NeurIPS Workshop, 2023.
>
> [6] Abels and Lenaerts. "Wisdom from diversity: bias mitigation through hybrid human-llm crowds." arXiv:2505.12349 (2025).
>
> > Q2: Why the focus on individual-level alignment?
>
> This is a valuable open question. While we cannot speak authoritatively for the entire community, we can offer a perspective from the LLM usage side.
>
> One likely contributing factor is that mainstream LLM agent personalization practices, e.g., from instruction-tuning to prompt design, are inherently **framed at the individual level**. Prompts typically follow patterns like "*You are an assistant who...*" or "*You are a young teacher who...*", instructing the model to **play an individual role**. This framing naturally directs attention toward individual-level alignment metrics.
>
> However, as extensive work in complex systems has demonstrated, individual-level alignment **does not guarantee collective-level alignment** (e.g., emergent misalignment [7]). This reinforces our position that social simulation research should prioritize collective-level validation rather than assuming that individual fidelity translates to population-level validity.
>
> We thank the reviewer for this thought-provoking question and will add a brief discussion of this issue in the revised manuscript.
>
> [7] Carichon et al. "The coming crisis of multi-agent misalignment: Ai alignment must be a dynamic and social process." arXiv:2506.01080 (2025).

---

> > ### Author Rebuttal · Reviewer_eXdr · 2026-04-02
> >
> > I thank the authors for their response, and am satisfied with the responses given in that it does clarify my doubts. I will maintain the positive review of the paper, and increase my confidence score to reflect my judgement accordingly.

---

### Official Review · Reviewer_D2oc · 2026-03-13

**Significance:** 1
**Argument Clarity:** 2
**Rating:** 3
**Confidence:** 4

**Questions:**

1.  This work mentions "explicitly reporting variance," but is there an emerging consensus on a "validity threshold"? At what point of variance divergence should a simulation result be considered "outside the boundary"?
2. If we use LLMs to generate synthetic populations to increase diversity, are we just witnessing a form of "autoregressive inbreeding" that masks the underlying lack of novel social insight?
3. Is the "Average Persona" mentioned in the paper inherently biased toward WEIRD (Western, Educated, Industrialized, Rich, and Democratic) populations? How should the "boundary" adapt when simulating non-Western social dynamics?

**Alternative Views Section:**

Yes

**Compliance With Llm Reviewing Policy A Conservative:**

Affirmed.

**Discussion Potential:**

2

**Final Justification:**

I still think some of my concerns only partially solved. But if this paper get accepted, I am fine with the decision.

**Paper Summary:**

This paper presents a critical position on the burgeoning field of LLM-based social simulations: they must be governed by clear "boundaries." The authors argue that because LLMs are fundamentally driven by likelihood-based loss functions, they tend to manifest an "Average Persona," leading to a significant loss of statistical heterogeneity.
The paper establishes a Mean-Variance Alignment Framework to diagnose simulation validity and conducts a systematic audit of 21 representative studies. The meta-analysis reveals a prevalent bias in current literature: researchers frequently validate "mean alignment" while neglecting "variance," often claiming excessive generalizability for simulations where heterogeneity is paramount. The authors conclude with a "Call to Action," proposing a set of boundary guidelines that match validation depth with the required degree of behavioral variance.

**Position:**

Yes

**Position In Title:**

Yes

**Related Work:**

2

**Strengths And Weaknesses:**

Strengths
1. In an era of unbridled optimism regarding LLM agents, this paper provides a necessary "cooling effect" by defining a scientific "boundary." Treating the "Average Persona" as a systematic architectural limitation rather than mere noise is a profound insight that challenges the current trajectory of Computational Social Science (CSS).
2. By simplifying the "validity" of human behavior into measurable statistical dimensions, the authors provide a practical roadmap for evaluating the claims of silicon-based societies.
3. This work provides a concrete set of guidelines. The recommendation to "match validation depth with the required variance" is particularly valuable for experimental design, ensuring that researchers do not overclaim the significance of qualitative collective patterns derived from low-variance agents.

Weaknesses
1. While the authors discuss the harm "Average Persona" does to statistical heterogeneity, the structural implications for marginalized groups deserve deeper exploration. If LLM simulations inherently gravitate toward the "distributional mean," minority perspectives and outlier behaviors are not just underrepresented—they are effectively erased. This "tyranny of the average" in silicon societies could lead to disastrously biased policy recommendations in the real world.
2. While the paper defines the "boundary," it is somewhat cautious in exploring how to "expand" it. Beyond prompt engineering and fine-tuning, are there more fundamental architectural interventions? For instance, could diversity-constrained decoding strategies or Mixture-of-Experts (MoE) models explicitly designed for demographic stratification serve as a bridge?
3. The paper identifies "optimists" and "skeptics," but the counter-arguments against the authors' position could be more robustly challenged. Specifically, the paper could benefit from a tighter logical deconstruction of why current alignment techniques (like RLHF) might actually exacerbate the "variance collapse" by further compressing the tail of the distribution.

**Support:**

2

---

> ### Author Rebuttal · Authors · 2026-03-29
>
> Thank you for your valuable feedback.
>
> > W1: Deeper exploration of implications for marginalized groups.
>
> We fully agree that "average persona" erasure goes beyond underrepresentation. Our manuscript addresses this at multiple points:
>
> - Section 4.1 (Lines 246–253) discusses subgroup heterogeneity erasure concentrating on dominant patterns reflecting social biases.
>
> - Section 4.2/Case 2 (Lines 280–286) discusses subgroup simulation biases and cultural deviations.
>
> - The Conclusion (Lines 431–436) emphasizes avoiding neglect of marginalized groups.
>
> - Appendix E (Lines 1014–1018) discusses bias and ethical concerns.
>
> We will strengthen the Conclusion section on this issue. We note that our statistical analysis of variance collapse and the reviewer's concern are complementary: documenting variance reduction is a necessary empirical basis for developing interventions ensuring equitable representation.
>
> > W2: Architectural interventions beyond prompt engineering and fine-tuning.
>
> We thank the reviewer for noting our discussion in Section 4.2 on Challenges in Enhancing Heterogeneity (Lines 289–314).  It was motivated by exploring whether existing methods can alleviate the boundary problems we identify. However, to the best of our knowledge, while existing approaches can ameliorate heterogeneity issues in specific scenarios, they have not yet been able to fully resolve the **fundamental lack of diversity** or the **systematic underrepresentation** of marginalized groups, as we describe in Section 4.2.
>
> Furthermore, upon more detailed investigation prompted by the reviewer's question, we find that even recent work (late 2025 – 2026) continues to identify **significant bottlenecks** in addressing complex human heterogeneity, cultural depth, and dynamic preferences through current intervention methods [1-3].
>
> [1] Prama et al. "Misalignment of LLM-Generated Personas with Human Perceptions in Low-Resource Settings." arXiv:2512.02058 (2025).
>
> [2] Venkit et al. "The Need for a Socially-Grounded Persona Framework for User Simulation." arXiv:2601.07110 (2026).
>
> [3] Tang et al. "PersonaFuse: A Personality Activation-Driven Framework for Enhancing Human-LLM Interactions." arXiv:2509.07370 (2025).
>
> This reinforces that a boundary-aware framework is a necessary prerequisite, not to prevent research, but to identify claim scopes that avoid overclaiming and inadvertent marginalization. We will acknowledge diversity-constrained decoding and MoE as intriguing future directions, while noting they remain largely unexplored for social simulation.
>
> > W3: Tighter critique of how RLHF may exacerbate variance collapse.
>
> We acknowledge that RLHF could further compress behavioral distributions. However, we chose not to develop detailed RLHF analysis because: (1) implementations vary substantially across models (e.g., PPO vs. DPO, different reward architectures), risking oversimplification; (2) our contribution is the application-level evaluation framework, not training procedure analysis.
>
> Importantly, our argument does not depend on the exact mechanism: our systematic review empirically demonstrates that LLM behaviors consistently show lower variance than human populations across domains, **regardless of technical origin**.
>
> Section 6 (Alternative Views) discusses why current alignment techniques do not resolve the alignment issues stated in our paper: LLM simulation proponents acknowledge diversity challenges (Anthis et al., 2025), and empirical evidence consistently shows distributional mismatches (Gao et al., 2025).
>
> > Q1: Is there an emerging consensus on a "validity threshold"?
>
> No consensus exists, partly because very few works have systematically examined variance boundaries, which is the gap we address.
>
> We discuss that variance should be assessed against available ground truth (Lines 231–237), with context-dependent thresholds: equilibrium studies may tolerate greater divergence than distributional or tipping-point studies (Appendix D).
>
> We advocate transparent reporting and context-appropriate evaluation rather than universal thresholds.
>
> > Q2: "Autoregressive inbreeding" in synthetic populations.
>
> We believe this concern directly aligns with our argument. Synthetic diversity from the same distributional space may merely re-sample within a constrained distribution. This is precisely why we argue ground truth comparison is essential (Section 5.2): synthetic diversity must be validated against empirical human data to distinguish genuine variation from autoregressive artifacts.
>
> > Q3: WEIRD bias in the "Average Persona".
>
> We agree. Section 4.2/Case 2 discusses how multilingual simulations show cultural inconsistencies (Jin et al., 2024; Naous & Xu, 2025).
>
> As noted in our response to W2, current interventions have not resolved culturally faithful simulation [1], further motivating boundary-aware evaluation with explicit cultural scoping of claims.

---

> > ### Author Rebuttal · Reviewer_D2oc · 2026-04-03
> >
> > There are some merits from the rebuttal, I would like to agree with part of the response.

---

### Official Review · Reviewer_QmCE · 2026-03-13

**Significance:** 3
**Argument Clarity:** 3
**Rating:** 4
**Confidence:** 4

**Questions:**

- The paper would benefit from discussing more concrete possible solutions or research directions for improving heterogeneity.

- It would also be helpful to briefly acknowledge that heterogeneity is not the only bottleneck in LLM based social simulation. Other important factors, such as temporal consistency, interaction structure, and environment design, may also strongly affect simulation validity.

- Given that the paper reviews 21 studies, it would be valuable to go one step further and synthesize recurring experimental design choices across these papers, especially those that may influence heterogeneity in practice.

**Alternative Views Section:**

Yes

**Compliance With Llm Reviewing Policy A Conservative:**

Affirmed.

**Discussion Potential:**

3

**Paper Summary:**

This paper argues that LLM based social simulations should be evaluated with clearer boundaries, especially when the research question depends on behavioral heterogeneity. The central concern is that LLMs may produce an “average persona” effect, where simulated agents can match human behavior in terms of mean responses while failing to reproduce the variance and diversity that many social phenomena depend on. To address this, the paper proposes a mean variance framework and reviews 21 recent LLM based social simulation studies to examine whether their validation practices are adequate for the level of heterogeneity their research questions require. The overall message is not that LLM based social simulation is invalid, but that claims should be matched more carefully to the degree of empirical support, especially when variance is crucial.

**Position:**

Yes

**Position In Title:**

Yes

**Related Work:**

3

**Strengths And Weaknesses:**

Strengths
- The paper raises a focused and important methodological concern. It identifies a specific issue, i.e., insufficient behavioral heterogeneity, and explains why this matters for many social science questions. The proposed mean variance framework is intuitive, since it makes clear that matching average human behavior is not enough when the target phenomenon depends on dispersion, extremes, or subgroup differences. Overall, the paper is careful in tone and makes a constructive case for boundary aware use of LLM based simulations.

Weaknesses
- While the paper acknowledges that experimental design choices such as prompting, persona construction, initial conditions, and model parameters can influence heterogeneity, it does not systematically extract or compare these design factors across the 21 reviewed studies.
- The paper is strong at setting boundaries and identifying risks, but less concrete about how researchers should improve heterogeneity or distinguish meaningful diversity from mere noise.

**Support:**

3

---

> ### Author Rebuttal · Authors · 2026-03-29
>
> Thank you for your insightful and constructive feedback.
>
> > W1/Q3: Lack of systematic extraction of design factors across the 21 studies.
>
> Thank you for this suggestion. Table 1, which reviews the 21 studies, was designed to evaluate validation practices, specifically whether researchers assess both mean alignment and variance against ground truth, and whether validation depth matches the heterogeneity demands of their research questions. Table 1 therefore focuses on **what is validated and how**, rather than on cataloguing experimental design choices. That said, we recognize that design choices can directly influence heterogeneity outcomes.
>
> To address your comments, we will revise the paper and add the following table documenting Persona Construction, Prompting Strategy, Model(s), Temperature, Number of Agents, and Interaction Structure for all these studies:
>
> [**Table 2 (external link)**](https://www.dropbox.com/scl/fi/lcqyh43ztjwpss3dxi84t/table2.pdf?rlkey=6sgyqzib98lxaywwyh5iyqe1g&st=qbb5i8xt&dl=0)
>
> A key finding from the above table is that lower-than-human variance appears across highly diverse design configurations, suggesting **no single design choice reliably explains variance outcomes**. Among the 9 papers that assessed variance, lower-than-human variance appeared across all design dimensions: diverse persona methods (from minimal to combined data driven + demographic + personality-based), multiple model families (GPT, Gemini, LLaMA, and Claude), temperatures from 0.0 to 1.2, and all interaction structures (pairwise, network, hybrid, and single-agent) at scales from 2 to 1M agents. No single design choice reliably distinguishes the two comparable-variance studies from the seven lower-variance ones. This cross-cutting pattern also supports our distinction between usage problems and boundary problems (Section 2.2): while design choices matter, they do not appear sufficient to overcome LLMs' inherent tendency toward behavioral homogeneity. The boundary-aware framework we propose remains **necessary regardless of design configuration**.
>
> > W2/Q1: More concrete solutions for improving heterogeneity.
>
> We appreciate this suggestion. Our Section 4.2 already discusses some challenges in enhancing heterogeneity and we will expand this with more directions from recent work:
>
> 1. **Data optimization:** Curating training corpora for social/cultural diversity, informed by sociolinguistic theories [1–3]. As recent work shows that purely synthetic data tends to suffer from diversity loss and long-tail collapse, carefully mixing real-world and synthetic data to preserve distributional tails may be a promising direction [4].
>
> 2. **Human feedback and fine-tuning:** Designing reward signals that value response diversity beyond average preferences [5–6].
>
> 3. **Crowd-based and hybrid aggregation:** Combining diverse human and LLM sources to mitigate bias [7].
>
> 4. **Multi-model ensembles:** Using multiple LLMs to increase population diversity, though even multi-LLM approaches may concentrate on few strategies (Fontana et al., 2024; Lu, 2024).
>
> [1] Yuan et al. "Measuring social norms of large language models." NAACL 2024 Findings.
>
> [2] Li et al. "From 1,000,000 users to every user: Scaling up personalized preference for user-level alignment." arXiv:2503.15463 (2025).
>
> [3] Grieve et al. "The sociolinguistic foundations of language modeling." Frontiers in Artificial Intelligence 7 (2025): 1472411.
>
> [4] Riaz et al. "MetaSynth: Meta-Prompting-Driven Agentic Scaffolds for Diverse Synthetic Data Generation." ACL 2025 Findings.
>
> [5] Achintalwar et al. "Alignment studio: Aligning large language models to particular contextual regulations." IEEE Internet Computing 28.5 (2024): 28-36.
>
> [6] Liu et al. "Trustworthy LLMs: a Survey and Guideline for Evaluating Large Language Models' Alignment." SoLaR, NeurIPS Workshop, 2023.
>
> [7] Abels and Lenaerts. "Wisdom from diversity: bias mitigation through hybrid human-llm crowds." arXiv:2505.12349 (2025).
>
> We note that these directions represent active areas of research, and none has yet been demonstrated to fully resolve the heterogeneity challenge. We cannot exhaustively discuss all possible approaches, but we believe these represent broadly recognized areas of concern.
>
> > Q2: Heterogeneity is not the only bottleneck.
>
> We fully agree with the reviewer. As noted in our manuscript, we discussed temporal consistency and robustness as additional boundary conditions in Appendix B. The reviewer's point that interaction structure and environment design also significantly affect simulation validity is well taken.
>
> In the revised manuscript, we will add a discussion of interaction structure (e.g., Sreedhar et al. (2025) mentioned in line 394) and environment design (e.g., action spaces that shape possible behaviors) in Appendix B, and clarify in the main text why heterogeneity is our chosen focus (Section 2.2).

---

### Official Review · Reviewer_DkUy · 2026-03-16

**Significance:** 4
**Argument Clarity:** 4
**Rating:** 5
**Confidence:** 3

**Questions:**

1. Is your usage of the term "boundary" with respect to LLMs standard in the social simulation community? My main concern is misinterpretation by the broader ML community, but if it is widely adopted by a different community, then it may be reasonable to continue using it.

**Alternative Views Section:**

Yes

**Compliance With Llm Reviewing Policy A Conservative:**

Affirmed.

**Discussion Potential:**

3

**Final Justification:**

My position on the paper has not changed after the rebuttal, as I did not have many concerns to be addressed. I still recommend the paper for acceptance.

**Paper Summary:**

The authors take the position that large language model (LLM)-based social simulations require clear boundaries in order to make any useful contributions to the social sciences. They point out a clear problem with current LLM-based social simulations: the LLM can replicate mean behavior over a population but is too homogeneous, not capturing the variance over the population. They further conduct a review of recent LLM-based social simulation studies to see how well their evaluation agrees with the demands of their research questions. Finally, they provide some heuristic boundaries and recommendations for LLM-based social simulations.

**Position:**

Yes

**Position In Title:**

Yes

**Related Work:**

4

**Strengths And Weaknesses:**

## Strengths
- Addresses a clear problem with LLM-based simulations in the poor alignment with real behavioral patterns, particularly due to lack of heterogeneity.
- The authors provide specific cases illustrating their claims in Section 4.2.
- Thorough literature survey with respect to their claims accompanying a call for future research.
- Very well written and a pleasure to read.

## Weaknesses
- The title is somewhat confusing. I was expecting the paper to be discussing something very different. I'm not sure how much of the ML community would associate boundaries in LLMs the way the authors are presenting. When I've heard the term "boundary" used with LLMs, it is typically to refer to "guardrails" or other safety mechanisms to prevent LLM hallucinations. Perhaps a longer title that includes words like "alignment" and "heterogeneity" might be useful.

**Support:**

4

---

> ### Author Rebuttal · Authors · 2026-03-29
>
> Thank you for your insightful and supportive review.
>
> > W1/Q1: The term "boundary" may cause confusion.
>
> We appreciate the reviewer's concern and agree that this is a valid point, especially given that the primary audience includes the broader ML community. We agree that in the ML context, "boundary" is more commonly associated with safety mechanisms or decision boundaries, which differs from our intended usage.
>
> We use "boundary" to refer to the methodological scope conditions of LLM-based social simulations, specifically, the limits of what claims can be reliably made given current LLM capabilities in generating behaviorally diverse agent populations. To address this, we will deliberate the title carefully and add an early definitional passage clarifying our use of the term boundary.

---

> > ### Author Rebuttal · Reviewer_DkUy · 2026-04-02
> >
> > I agree with the authors' proposed approach to handle the potential confusion of the term "boundary" and continue to support the paper for acceptance.

---

### Decision · Program_Chairs · 2026-04-30

**Decision:**

Accept (regular)

**Comment:**

No serious issues emerged in the discussion with the reviewers. I do think a somewhat important omission is that relatively little space is devoted to cases where the social simulation apparently worked out well e.g Argyle et al. (2023), the paper is mentioned but not really engaged with. The "algorithmic fidelity" concept developed in that paper would be a useful concept to discuss here since it's a validity concept for LLM-based social simulation, but it is not mentioned.  I don't think mentioning it would undermine anything about the broader claim, but it really should be mentioned. Anyway, I don't think this omission is serious enough to sink the paper, but I do hope the authors will consider adding more discussion on this as they revise for the camera ready phase.